# Extremely Preterm Babies—Legal Aspects and Palliative Care at the Border of Viability

**DOI:** 10.3390/children9101594

**Published:** 2022-10-21

**Authors:** Reinhard Dettmeyer

**Affiliations:** Institute of Forensic Medicine, Justus-Liebig-University Giessen, Frankfurter Str. 58, D-35392 Gießen, Germany; reinhard.dettmeyer@forens.med.uni-giessen.de; Tel.: +49-(0)-641-99-41411; Fax: +49-(0)-641-99-41419

**Keywords:** extremely preterm babies, viability, palliative care, rights of parents, court decision

## Abstract

There are various legal considerations and rare decisions of courts in western countries concerning palliative care and the border of viability in cases of extremely preterm babies. Nevertheless, on the one hand, regulations and decisions of courts describe the conditions physicians have to accept. On the other hand, courts are also able to accept that every case can be special, and needs a unique answer. Therefore, the framework can be described as well as the medical disciplines, which should be involved in a particular case.

## 1. Introduction

Perinatal care at the borderline of viability leads to several medicolegal aspects [1], not only questions of informed consent, but also, for example, whether resuscitation is a real option in the management of babies born extremely preterm at less than 26 weeks of gestation [2]. The predicted outcome in such cases must be known, with differences between regions and developed or less developed countries [3,4,5,6,7,8], and, of course, with regard to the single newborn concerned, before end-of-life decisions for extremely low-gestational-age infants are in discussion. Obviously, it is difficult to establish simple rules for complicated decisions [9]. 

There are various legal considerations and rare decisions of courts in western countries concerning palliative care and the border of viability in cases of extremely preterm babies. In Germany, decisions of courts are also very rare as far as extremely preterm babies are concerned. Nevertheless, questions of limitations in the treatment of extremely preterm babies with fatal prognoses need an answer. As far as legal aspects are concerned, such answers are primarily based on general regulations as the constitutional law with regard to a widely accepted ethical background. 

Nevertheless, the right of parents to obtain all the information necessary to decide in the child’s best interest must be taken into account, and the religion of the parents could be of special interest, because decisions based on religious beliefs can also be protected by the constitution [10]. Additionally, therapeutic options offered by specialized physicians, such as neonatologists or physicians of other medical specialties, are always of interest to make decisions in a single case. Guidelines, e.g., from the Association of the Scientific Medical Societies, although not mandatory, help to define the expected medical standard [11,12]. It should also be mentioned that only countries with a highly developed healthcare system provide neonatal-patient-centered care, giving the necessary attention to extremely preterm babies. Therefore, in most countries, there is no real option to treat such babies. Decisions of courts are rare and derive not from courts in countries with a less-developed medical service. 

### 1.1. Legal Aspects 

In general, decisions on measures to save the lives of extremely preterm babies undergo the same judicial rules and ethic aspects compared to other medical decisions where the “futility” of medical help is in doubt [13]. Every child has the right to receive individual treatment independent from the prognosis of survival. There are rights of the child written down in the “Convention on the Rights of the Child, CRC” of the United Nations, also in Germany [14]. Physicians have to observe the dignity of the patient (Basic Constitution Law of the Federal Republic of Germany, Article 1) as well as the right to live and the right of physical integrity (Basic Constitution Law of the Federal Republic of Germany, Article 2). Nevertheless, physicians do not have to preserve lives in any circumstances. However, every child must receive basic services, alleviation of pain, best care, and human attention [15]. Indicated measures need to be agreed by the patient or the parents of the preterm baby [16]. These parents of babies born severely premature or with serious abnormalities are rarely turning to the courts in a bid to override medical opinion to commence or continue life-sustaining treatment. However, decisions are needed if extremely preterm babies with severe and untreatable diseases develop a fatal prognosis. Dependent from the local situation, rules fixed in the Medical Association’s professional code of conduct can also be relevant, rarely special medicolegal laws such as the “Born-Alive Infants Protection Act of 2001” in the USA [17]. 

There is no special written law in Germany or most other countries concerning the obligation to start medical treatment in cases of preterm babies. Except rare cases, up to now there is no real chance of survival when born with a gestational age of less than 22 weeks. This means, there is no obligation to treat a preterm baby when born earlier than 22 weeks (Japan, Austria, Italy), 23 weeks (USA, UK) or 24 weeks (Switzerland, Netherland, France). In accordance, a German district court also did not see any obligation to treat a preterm baby born in week 23 [18]. Therefore this discussion concerns the situation of babies when born between 22 and 24 weeks.

In Germany, while there is no obligation to treat preterm babies when born earlier than 22 weeks, a single decision of a court is known and medical guidelines, although not mandatory, have to be taken into account. Both, guidelines and court decisions firstly ask for informed consent. This means an agreement between parents and caregivers if intense measures will not improve the chance of survival. If they pose an unacceptable burden to the child, those measures can be withheld. A so-called “do not resuscitate order (DNR)”, to handle the situation after birth or later in case an acute heart arrest occurs, can be helpful in cases without any chance to survive longer than minutes, hours, or days. Questions of limitations in the treatment arise and an informed consent with the parents must be found. 

There is no option for decisions before the infant is born. On the one hand, as stated by a court in the USA, a fully informed decision could not be made until the birth of the infant, and, at that time, neonatologists are faced with “emergent circumstances” [19]. On the other hand, the birth of an extremely preterm baby always leads to an emergent situation. 

In Australia, a court was asked for a decision for the first time in 2011 concerning a preterm baby undergoing palliative care (Case of Baby D.). The judge asked for a decision accepted by both the physicians and the parents. If achieved, there will be no prosecution, even in case the decision will lead to death. Courts in Australia also accepted to stop the treatment of a nine-month-old boy with several severe diseases and artificial respiration, although parents insisted to continue all measures to save the life (Case of Mohammed). A court in Texas, USA, enables physicians to resuscitate a potentially viable, extremely preterm baby against the declared intention of the parents (Case of Kara Miller, gestation age 23.1 weeks, 629 g; Miller vs. Health Care Corporation; HCA). Other courts in Wisconsin, USA, and Washington, USA, also decided physicians must not resuscitate in every case parents ask for all medical measures (Montalvo vs. Borkovec, 2002, Wis. App. 147–Ct. App. Wis. 2002; Stewart-Graves vs. Vaughn, 170 P.3d 1151), based on a former decision made in 1994, where a court did not declare manslaughter in the case a father took his child away from life-sustaining measures (Case of People vs. Messenger, 221 Michigan, App. 171–Mich. Ct. App. 1997, USA). 

### 1.2. Conflicts

Different situations can lead to severe conflicts and should be regarded separately: 

#### 1.2.1. There Is No Time to Care for Informed Consent

Decisions concerning life-saving measures before birth obviously will not be accepted by courts because nobody can see whether there will be an emergent situation or not (Supreme Court of Texas). If there is no time to ask for informed consent, physicians are allowed to decide. The Washington Supreme Court decided that informed consent was not required if the infant would have died if resuscitation was delayed to obtain consent: “Such a decision cannot be truly “informed” … when the circumstances permit no more than a hasty explanation of probable outcomes by a physician whose attention must primarily focus on lifesaving efforts” (Stewart-Graves vs. Vaughn, 170 P.3d 1151, Wash. 2007).

#### 1.2.2. Physicians Want to Start Therapy, Parents Refuse to Agree, Partly Because of Seeing a Long-Term Morbidity and Suffering of the Child

Although child custody is part of the rights of the parents to decide in the best interest of their newborn, German law did not allow decisions leading to death when there is a real chance of the newborn to survive according to the medical assessment. This assessment can be uncertain, but physicians must use every chance to save life, independent how small hope is. The 2003-decision of the Texas Supreme Court in the case of Sidney Miller, a 615-g, 23-week-gestation preterm infant born in 1990 and resuscitated against the wishes of the parents, did accept the decision of the physicians to resuscitate (Miller vs. HCA, 118S.W. 3d 758, Texas 2003). 

This means that if parents refuse to agree to the physicians suggested therapy, the physicians should go to court and medical child custody will be withdrawn from the parents and transferred to a caregiver. Treating physicians should not accept the role of a caregiver in addition, because conflicting interests should be avoided. Nevertheless, the described situation concerning an extremely preterm baby can also be happen concerning babies born later and with severe diseases. In case parents refuse to accept the suggested therapy in all these and similar cases, the court should make a decision. 

#### 1.2.3. Physicians Cannot Offer a Therapy and Prefer Palliative Care, Nevertheless, Parents Ask for Maximal Therapy

To negate any therapeutic options includes the ascertainment that there is no chance for the newborn to survive more than a very, very short time. If so, parents, although having child custody, cannot enforce any treatment except measures as part of palliative care. Physicians have to inform the parents very carefully with regard to the newborns situation and about all therapeutic options and the negative perspective. It is also acceptable to inform the parents about available data on survival rates and outcomes of extremely low gestational infants (22–25 week’s gestation), although these data present with wide variations from country to country [19,20]. Additionally, available guidelines from different countries show a greater variation concerning newborns between 23 and 24 weeks of gestation. Some of them support active care not earlier than 25 weeks of gestation. De jure physicians have to inform the parents and treat newborns according to the medical standards of their country and their hospital. 

If, finally, the parents do not agree, they are not embarrassed to go to court. Therefore, in such cases, a clinical ethic committee can be involved to discuss the situation, to document a statement and to support physicians. This way to manage the situation by involving the ethic committee can help to prevent allegations. Despite the decision to start a course of therapy or not, additional consent must be provided regarding resuscitation, if necessary. 

## 2. Perspectives and Conclusions

Courts, if necessary, decide single cases, but each time with regard to the actual development of medical standards. Therefore, the border of 22–25 weeks of gestation is not fixed forever. If mortality rates of extremely low-birth-weight infants demonstrate definite improvement, courts will change their opinions, advised by medical experts. In this context, medical guidelines are of general interest to define medical standards, but they are not mandatory, unlike laws. With regard to the daily work, decisions of the physicians should be accepted by the parents; nevertheless, parents can ask for maximal therapy, on the one hand, and have to decide about resuscitation, if necessary, suddenly and more or less unexpectedly, on the other hand. 

To provide a framework for decisions can be helpful. Decisions should be made by physicians of several specialties, the nursing staff should be included, a neutral ethics committee with juristic expertise should be asked, and the parents must always be involved step-by-step on the way to a decision. Every step should be documented carefully. Considering these guidelines, appeal to the court should be avoidable in most cases.

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
