# Peer review of "Extremely Preterm Babies—Legal Aspects and Palliative Care at the Border of Viability"

_children, 2022, doi:10.3390/children9101594_

Round 1
Reviewer 1 Report
I feel the paper is important but needs improved flow, direction, and conclusion. It is hard to follow the goals of the authors in the paper, other than being born between 22-24 weeks can lead to different goals between parents and physicians.
Author Response
Indeed, other than being born between 22-24 weeks can lead to different goals between parents and physicians. But the article just focuses on 22-24 weeks and I think, this is understandable to the readers.
Reviewer 2 Report
This is an interesting review regarding the care of infants at the border of viability. There are some issues that should be improved:
- I think it is more appropriate to use the word “infants” as opposed to “babies” in scientific/medical papers, so I recommend to the author to replace this word throughout
- I’m not sure why this manuscript doesn’t have row numbers, it is certainly more difficult to review…
- In the Introduction section, there are some ambiguous phrases, such as: “And, of course, with regard to the single newborn concerned before End-of-life decisions for extremely low-gestational-age infants are in discussion.”, also “Decisions of courts are rare and derive from courts in less countries, as far as available”. Both of these should be rephrased, as they lack meaning in their current form
- The author mentions in the Introduction section the fact that resuscitation below 26 weeks is often not an option. I recommend to the author to change this limit to 25 weeks, simply because nowadays, all over the world an infant of 25 gestational weeks (usually 700-800 grams) will be resuscitated. The main issue around the border of viability argument stems at 22 to 24 weeks – it is in fact reiterated on the second page of the manuscript. Also in the same phrase, the predicted outcome of the infant will be different inside the same region/country, depending on the experience and assigned level of care of each hospital – in theory, each hospital should have their own statistics regarding the major morbidities they treat
- In the Legal aspects section, especially towards the end of it, the author seems to be mixing up the two situations – (1) withholding active resuscitation and intensive care in the delivery room/operating theater based on the limit of viability or “emergent situations” and (2) withdrawing the provision of intensive care after an initial period of doing so. There should be a differentiation between the two situations
- The Conflicts section is fairly well written. I only have an issue with the word “embarrassed” when referring to parents, on the first row of the last paragraph. In my opinion, that’s not the best word used in this situation and should be swapped
- The last section should be numbered as 3. The borderline of viability should again be 22-24 weeks.
- There should be a thorough check of English phrasing throughout the manuscript.
Author Response
According to this review, the word "babies" is replaced and the word "infants" inserted.
One phrase in the Introduction section is "ambiguous" as written in the review. The word "before" must be replaced. But the other phrase, mentioned in the review, is a simple statement, that decisions of courts are rare and derive from courts in less countries (that´s as it is).
Indeed, meanwhile resuscitation below 25 weeks should be written.
The Legal aspects section was controlled once more, especially the end of it.
But there, the authors presents decisions of courts and therefore, if so, it is not the author who seems to be mixing up two situations, that´s the problem of the decisions of courts.
The word "embarrassed" is replaced and "not detained" is written.
Of course, the last section should be numbered as 3 and the borderline of viability should again be 22-24 weeks.
Round 2
Reviewer 1 Report
The introduction and overall manuscript does not read well.
Author Response
I already added some more text, I have made minor corrections, e.g. "infants" instead of "babies".